# Gradient Learning under Tilted Empirical Risk Minimization

**DOI:** 10.3390/e24070956

**Published:** 2022-07-09

**Authors:** Liyuan Liu, Biqin Song, Zhibin Pan, Chuanwu Yang, Chi Xiao, Weifu Li

**Affiliations:** 1College of Science, Huazhong Agricultural University, Wuhan 430062, China; liulymt@foxmail.com (L.L.); biqin.song@mail.hzau.edu.cn (B.S.); pzbhallow@mail.hzau.edu.cn (Z.P.); 2Hubei Key Laboratory of Applied Mathematics, Hubei University, Wuhan 430062, China; 3School of Electronic Information and Communications, Huazhong University of Science and Technology, Wuhan 430074, China; chuanwuyang@hust.edu.cn; 4Key Laboratory of Biomedical Engineering of Hainan Province, School of Biomedical Engineering, Hainan University, Haikou 570228, China

**Keywords:** gradient learning, operator approximation, reproducing kernel Hilbert spaces, tilted empirical risk minimization

## Abstract

Gradient Learning (GL), aiming to estimate the gradient of target function, has attracted much attention in variable selection problems due to its mild structure requirements and wide applicability. Despite rapid progress, the majority of the existing GL works are based on the empirical risk minimization (ERM) principle, which may face the degraded performance under complex data environment, e.g., non-Gaussian noise. To alleviate this sensitiveness, we propose a new GL model with the help of the tilted ERM criterion, and establish its theoretical support from the function approximation viewpoint. Specifically, the operator approximation technique plays the crucial role in our analysis. To solve the proposed learning objective, a gradient descent method is proposed, and the convergence analysis is provided. Finally, simulated experimental results validate the effectiveness of our approach when the input variables are correlated.

## 1. Introduction

Data-driven variable selection aims to select informative features related with the response in high-dimensional statistics and plays a critical role in many areas. For example, if the milk production of dairy cows can be predicted by the blood biochemical indexes, then the doctors are eager to know which indexes can drive the milk production because each of them is independently measured with additional burden. Therefore, an explainable and interpretable system to select the effective variables is critical to convince domain experts. Currently, the methodologies on variable selection methods can be roughly divided into three categories including linear models [1,2,3], nonlinear additive models [4,5,6], and partial linear models [7,8,9]. Although achieving promising performance in some applications, these methods mentioned above still suffer from two main limitations. Firstly, the target function of these methods is restricted on the assumption of specific structures. Secondly, these methods cannot revive how the coordinates vary with respect to each other. As an alternative, Mukherjee and Zhou [10] proposed the gradient learning (GL) model, which aims to learn the gradient functions and enjoys the model-free property.

Despite the empirical success [11,12,13], there are still some limitations of the GL model, such as high computational cost, lacking the sparsity in high-dimensional data and lacking the robustness to complex noises. To this end, several variants of the GL model have been devoted to developing alternatives for individual purposes. For example, Dong and Zhou [14] proposed a stochastic gradient descent algorithm for learning the gradient and demonstrated that the gradient estimated by the algorithm converges to the true gradient. Mukherjee et al. [15] provided an algorithm to reduce dimension on manifolds for high-dimensional data with few observations. They obtained generalization error bounds of the gradient estimates and revealed that the convergence rate depends on the intrinsic dimension of the manifold. Borkar et al. [16] combined ideas from Spall’s Simultaneous Perturbation Stochastic Approximation with compressive sensing and proposed to learn the gradient with few function evaluations. Ye et al. [17] originally proposed a sparse GL model to further address the sparsity for high-dimensional variable selection of the estimated sparse gradients. He et al. [18] developed a three-step sparse GL method which allows for efficient computation, admits general predictor effects, and attains desirable asymptotic sparsistency. Following the research direction of robustness, Guinney et al. [19] provided a multi-task model which are efficient and robust for high-dimensional data. In addition, Feng et al. [20] provided a robust gradient learning (RGL) framework by introducing a robust regression loss function. Meanwhile, a simple computational algorithm based on gradient descent was provided, and the convergence of the proposed method is also analyzed.

Despite rapid progress, the GL model and its extensions mentioned above are established under the framework of empirical risk minimization (ERM). While enjoying the nice statistical properties, ERM usually performs poorly in situations where average performance is not an appropriate surrogate for the problem of interest [21]. Recently, a novel framework, named tilted empirical risk minimization (TERM), is proposed to flexibly address the deficiencies in ERM [21]. By using a new loss named *t*-tilted loss, it has been shown that TERM (1) can increase or decrease the influence of outliers, respectively, to enable fairness or robustness; (2) has variance reduction properties that can benefit generalization; and (3) can be viewed as a smooth approximation to a superquantile method. Considering these strength, we propose to investigate the GL under the framework of TERM. The main contributions of this paper can be summarized as follows:New learning objective. We propose to learn the gradient function under the framework of TERM. Specifically, the *t*-tilted loss is embedded into the GL model. To the best of our knowledge, it may be the first endeavor in this topic.Theoretical guarantees. For the new learning objective, we estimate the generalization bound by error decomposition and operator approximation technique, and further provide the theoretical consistency and the convergence rate. To be specific, the convergence rate can recover the result of traditional GL as *t* tends 0 [10].Efficient computation. A gradient descent method is provided to solve the proposed learning objective. By showing the smoothness and strongly convex of the learning objective, the convergence to the optimal solution is proved.

The rest of this paper is organized as follows: Section 2 proposes the GL with *t*-tilted loss (TGL) and states the main theoretical results on the asymptotic estimation. Section 3 provides the computational algorithm and its convergence analysis. Numerical experiments on synthetic data sets will be implemented in Section 4. Finally, Section 5 closes this paper with some conclusions.

## 2. Learning Objective

In this section, we introduce TGL and provide the main theoretical results on the asymptotic estimation.

### 2.1. Gradient Learning with *t*-Tilted Loss

Let *X* be a compact subset of $R^{n}$ and Y∈R. Assume that ρ is a probability measure on Z:=X×Y. It induces the marginal distribution $\rho_{X}$ on *X* and conditional distributions ρ(·|x) at x∈X. Denote $L_{\rho_{X}}^{2}$ as the $L^{2}$ space with the metric ${∥f∥}_{\rho }=(\int_{X}{\left|f\left(x\right)\right|}^{2}d\;\rho_{X}{)}^{1/2}$. In addition, the regression function $f_{\rho }:X\to Y$ associated with ρ is defined as
$$f_{\rho }\left(x\right)=\int_{Y}yd\;\rho \left(y|x\right),\;\;x\in X.$$

For $x={\left(x^{1},x^{2},\cdots ,x^{n}\right)}^{T}\in X$, the gradient of $f_{\rho }$ is the vector of functions (if the partial derivatives exist)
$$\nabla f_{\rho }={\left(\frac{\partial f_{\rho }}{\partial x^{1}},\frac{\partial f_{\rho }}{\partial x^{2}},\ldots ,\frac{\partial f_{\rho }}{\partial x^{n}}\right)}^{T}.$$

The relevance between the *l*-th coordinate and $f_{\rho }$ can be evaluated via the norm of its partial derivative $∥\frac{\partial f_{\rho }}{\partial x^{l}}∥$, where a large value implies a large change in the function $f_{\rho }$ with respect to a sensitive change in the *l*-th coordinate. This fact gives an intuitive motivation for the GL. In terms of Taylor series expansion, the following equation holds:(1)$$f_{\rho }\left(x\right)\approx f_{\rho }\left(\tilde{x}\right)+\nabla f_{\rho }\left(\tilde{x}\right)\cdot \left(x-\tilde{x}\right),$$
for $x\approx \tilde{x}$ and $x,\tilde{x}\in X$. Inspired by (Equation 1), we denote the weighted square loss of $\vec{f}$ as
(2)$$V(\vec{f},z,\tilde{z}=\omega \left(x,\tilde{x}\right){\left(\tilde{y}-y+\vec{f}{\left(\tilde{x}\right)}^{T}\left(x-\tilde{x}\right)\right)}^{2},\;\;\;\vec{f}\in {\left(L_{\rho_{X}}^{2}\right)}^{n},\;z,\tilde{z}\in Z,$$
where the restriction $x\approx \tilde{x}$ will be enforced by weights $\omega (x,\tilde{x})$ given by $\frac{1}{s^{n+2}}e^{-|x-\tilde{x}{|}^{2}/2s^{2}}$ with a constant 0<s≤1, see, e.g., [10,11,19]. Then, the expected risk of $\vec{f}$ can be given by
(3)$$E\left(\vec{f}\right)=\int_{Z}\int_{Z}V\left(\vec{f},z,\tilde{z}\right)d\;\rho \left(z\right)d\;\rho \left(\tilde{z}\right).$$

As mentioned in [21], the $\vec{f}$ defined in (Equation 3) usually performs poorly in situations where average performance is not an appropriate surrogate. Inspired from [21], for $t\in R^{\setminus 0}$, we address the deficiencies by introducing the *t*-tilted loss and define the expected risk of $\vec{f}$ with *t*-tilted loss as
(4)$$E\left(\vec{f},t\right)=\frac{1}{t}\log\int_{Z}\int_{Z}e^{tV(\vec{f},z,\tilde{z})}d\;\rho \left(z\right)d\;\rho \left(\tilde{z}\right).$$

**Remark** **1.**
*Note that $t\in R^{\setminus 0}$ is a real-valued hyperparameter, and it can encompass a family of objectives which can address the fairness (t>0) or robustness (t<0) by different choices. In particular, it recovers the expected risk (Equation 3) as t→0.*


On this basis, the GL with *t*-tilted loss is formulated as the following regularization scheme:(5)$${\vec{f}}_{\lambda ,t}=\underset{\vec{f}\in H_{K}^{n}}{\arg\min}\{E\left(\vec{f},t\right)+\lambda ∥\vec{f}{∥}_{K}^{2}\},$$
where λ>0 is a regularization parameter. Here, K:X×X→R is a Mercer kernel that is continuous, symmetric, and positive semidefinite [22,23] and $H_{K}$ induced by *K* be an RKHS defined as the closure of the linear span of the set of functions $\left\{K_{x}:=K\left(x,\cdot \right):x\in X\right\}$ with the inner product ${\langle \cdot ,\cdot \rangle }_{K}$ satisfying ${\langle K_{x},K_{\tilde{x}}\rangle }_{K}=K\left(x,\tilde{x}\right)$. The reproducing property takes the form ${\langle K_{x},f\rangle }_{K}=f\left(x\right),\;\forall x\in X,\;\forall f\in H_{K}.$ Then, we denote $H_{K}^{n}$ as an *n*-fold RKHS with the inner product
$$\begin{array}{c} {\langle \vec{f},\vec{h}\rangle }_{K}=\sum_{l=1}^{n}{\langle f^{l},h^{l}\rangle }_{K},\;\;\vec{f}={\left(f^{1},f^{2},\ldots ,f^{n}\right)}^{T},\;\;\vec{h}={\left(h^{1},h^{2},\ldots ,h^{n}\right)}^{T}\in H_{K}^{n}, \end{array}$$
and norm $∥\vec{f}{∥}_{K}^{2}={\langle \vec{f},\vec{f}\rangle }_{K}$.

### 2.2. Main Results

This subsection states our main theoretical results on the asymptotic estimation of $∥{\vec{f}}_{\lambda ,t}-\nabla f_{\rho }{∥}_{\rho }$ on the space ${\left(L_{\rho_{X}}^{2}\right)}^{n}$ with norm $∥\vec{f}{∥}_{\rho }=(\sum_{l=1}^{n}∥f^{l}{{∥}_{\rho }^{2})}^{1/2}$. Before proceeding, we provide some necessary assumptions which have been used extensively in machine learning literature, e.g., [24,25].

**Assumption** **1.**
*Supposing that $\nabla f_{\rho }\in H_{K}^{n}$ and the kernel K is $C^{3}$, there exists a constant $c_{\upsilon }>0$ such that*

(6)
$$|f_{\rho }\left(x\right)-f_{\rho }\left(\tilde{x}\right)-\nabla f_{\rho }{\left(\tilde{x}\right)}^{T}\left(x-\tilde{x}\right)|\leq c_{\upsilon }{\left|x-\tilde{x}\right|}^{2},\;\;\;\;\forall x,\tilde{x}\in X.$$



**Assumption** **2.**
*Assume |y|≤M, $\left|x\right|\leq M_{X}$ almost surely. Suppose that, for some $\varsigma \in (0,\frac{2}{3})$, $c_{l},c_{h}>0$, the marginal distribution $\rho_{X}$ satisfies*

(7)
$$\rho_{X}\left(\left\{x\in X:\underset{\tilde{x}\in R^{n}\setminus X}{\inf}\left|x-\tilde{x}\right|\leq s\right\}\right)\leq c_{h}^{2}s^{4\varsigma },\;\;\;\;\forall s>0,$$

*and the density p(z) of dρ(z) exists and satisfies*

(8)
$$c_{l}\leq p\left(z\right)\leq c_{h},\;\;\;\;\left|p\left(z\right)-p\left(\tilde{z}\right)\right|\leq c_{h}{\left|z-\tilde{z}\right|}^{\varsigma },\;\;\;\;\forall z,\tilde{z}\in Z.$$



Taking the functional derivatives of (Equation 5), we know that ${\vec{f}}_{\lambda ,t}$ can be expressed in terms of the following integral operator on the space ${\left(L_{\rho_{X}}^{2}\right)}^{n}$.

**Definition** **1.**
*Let integral operator $L_{K,s}:{\left(L_{\rho_{X}}^{2}\right)}^{n}\to {\left(L_{\rho_{X}}^{2}\right)}^{n}$ be defined by*

(9)
$$\begin{array}{c} L_{K,s}\vec{f}=\int_{Z}\int_{Z}\phi \left(z,\tilde{z}\right)\omega \left(x,\tilde{x}\right)\left(\vec{f}{\left(\tilde{x}\right)}^{T}\left(x-\tilde{x}\right)\right)K_{\tilde{x}}\left(x-\tilde{x}\right)d\;\rho \left(\tilde{z}\right)d\;\rho \left(z\right), \end{array}$$

*where*

$$\phi \left(z,\tilde{z}\right)={\left(\int_{Z}\int_{Z}e^{tV({\vec{f}}_{\lambda ,t},u,v)}d\;\rho \left(u\right)d\;\rho \left(v\right)\right)}^{-1}e^{tV({\vec{f}}_{\lambda ,t},z,\tilde{z})}.$$



The operator $L_{K,s}$ has its range in $H_{K}^{n}$. It can also be regarded as a positive operator on $H_{K}^{n}$. We shall use the same notion for the operators on these two different domains. Given the definition of integral operator $L_{K,s}$, we can write ${\vec{f}}_{\lambda ,t}$ in the following equation.

**Theorem** **1.**
*Given the integral operator $L_{K,s}$, we have the following relationship:*

(10)
$$\begin{array}{c} {\vec{f}}_{\lambda ,t}={\left(L_{K,s}+\lambda I\right)}^{-1}{\vec{f}}_{\rho ,s}, \end{array}$$

*where ${\vec{f}}_{\rho ,s}=\int_{Z}\int_{Z}\phi \left(z,\tilde{z}\right)\omega \left(x,\tilde{x}\right)\left(f_{\rho }\left(x\right)-f_{\rho }\left(\tilde{x}\right)\right)K_{\tilde{x}}\left(x-\tilde{x}\right)d\;\rho \left(\tilde{z}\right)d\;\rho \left(z\right)$, and I is the identity operator.*


**Proof of Theorem** **1.**To solve the scheme (Equation 5), we take the functional derivative with respect to $\vec{f}$, apply it to an element $\delta \vec{f}$ of $H_{K}^{n}$ and set it equal to 0. We obtain
$$\begin{array}{c} \int_{Z}\int_{Z}\phi \left(z,\tilde{z}\right)\omega \left(x,\tilde{x}\right)\left(\tilde{y}-y+{\vec{f}}_{\lambda ,t}{\left(\tilde{x}\right)}^{T}\left(x-\tilde{x}\right)\right)\delta \vec{f}{\left(\tilde{x}\right)}^{T}\left(x-\tilde{x}\right)d\;\rho \left(\tilde{z}\right)d\;\rho \left(z\right)+\lambda {\langle {\vec{f}}_{\lambda ,t},\delta \vec{f}\rangle }_{K}=0. \end{array}$$Since it holds for any $\delta \vec{f}\in H_{K}^{n}$, it is trivial to obtain
$$\begin{array}{c} \int_{Z}\int_{Z}\phi \left(z,\tilde{z}\right)\omega \left(x,\tilde{x}\right)\left(\tilde{y}-y+{\vec{f}}_{\lambda ,t}{\left(\tilde{x}\right)}^{T}\left(x-\tilde{x}\right)\right)K_{\tilde{x}}\left(x-\tilde{x}\right)d\;\rho \left(\tilde{z}\right)d\;\rho \left(z\right)+\lambda {\vec{f}}_{\lambda ,t}=0 \end{array}$$
and
$$\begin{array}{c} \lambda {\vec{f}}_{\lambda ,t}+L_{K,s}{\vec{f}}_{\lambda ,t}={\vec{f}}_{\rho ,s}. \end{array}$$The desired result follows by shifting items.    □

On this basis, we propose to bound the error $∥{\vec{f}}_{\lambda ,t}-\nabla f_{\rho }{∥}_{\rho }$ by a functional analysis approach and present the error decomposition as following proposition. The proof is straightforward and omitted for brevity.

**Proposition** **1.**
*For the ${\vec{f}}_{\lambda ,t}$ defined in (Equation 5), it holds that*

(11)
$$∥{\vec{f}}_{\lambda ,t}-\nabla f_{\rho }{∥}_{\rho }\leq ∥{\vec{f}}_{\lambda ,t}-\nabla f_{\rho }+\lambda {\left(L_{K,s}+\lambda I\right)}^{-1}\nabla f_{\rho }{∥}_{\rho }+{∥\lambda {\left(L_{K,s}+\lambda I\right)}^{-1}\nabla f_{\rho }∥}_{\rho }.$$



In the sequel, we focus on bounding $∥{\vec{f}}_{\lambda ,t}-\nabla f_{\rho }+\lambda {\left(L_{K,s}+\lambda I\right)}^{-1}\nabla f_{\rho }{∥}_{\rho }$ and $∥\lambda {\left(L_{K,s}+\lambda I\right)}^{-1}\nabla f_{\rho }{∥}_{\rho }$, respectively. Before we embark on the proof, we single out a important property regarding $\phi (z,\tilde{z})$ that will be useful in later proofs.

**Lemma** **1.**
*Under the Assumptions 1 and 2, there exists $B_{t}$ and $A_{t}$ dependent on t satisfying*

(12)
$$B_{t}=e^{-8|t|(M^{2}+C_{K}M_{X})}\leq \phi \left(z,\tilde{z}\right)\leq A_{t}=e^{8|t|(M^{2}+C_{K}M_{X})}.$$



**Proof of Lemma** **1.**Since the kernel *K* is $C^{3}$ and ${\vec{f}}_{\lambda ,t}\in H_{K}^{n}$, we know from Zhou [26] that $f_{\lambda ,t}^{l}$ is $C^{1}$ for each *l*. There exists a constant $C_{K}$ satisfying $|{\vec{f}}_{\lambda ,t}{\left(x\right)|}^{2}\leq C_{K},\forall x\in X$. Hence, using Cauchy inequality, we have
$$\begin{array}{cc} V({\vec{f}}_{\lambda ,t},z,\tilde{z}) & =\omega \left(\tilde{x},x\right){\left(\tilde{y}-y+{\vec{f}}_{\lambda ,t}{\left(\tilde{x}\right)}^{T}\left(x-\tilde{x}\right)\right)}^{2}\\ & \leq 2(4M^{2}+|{\vec{f}}_{\lambda ,t}\left(\tilde{x}\right){|}^{2}|x-\tilde{x}{|}^{2})\\ & \leq 8(M^{2}+C_{K}M_{X}). \end{array}$$By a direct computation, we obtain
$$\begin{array}{c} e^{-8|t|(M^{2}+C_{K}M_{X})}\leq {\left(\int_{Z}\int_{Z}e^{tV({\vec{f}}_{\lambda ,t},u,v)}d\;\rho \left(u\right)d\;\rho \left(v\right)\right)}^{-1}e^{tV({\vec{f}}_{\lambda ,t},z,\tilde{z})}\leq e^{8|t|(M^{2}+C_{K}M_{X})}. \end{array}$$The desired result follows.    □

Denote $\kappa =\sup_{x\in X}K\left(x,x\right)$ and the moments of the Gaussian as $J_{p}=\int_{R^{n}}e^{-\frac{{\left|x\right|}^{2}}{2}}{\left|x\right|}^{p}d\;x$, p=1,2,3,…, we establish the following Lemma.

**Lemma** **2.**
*Under Assumptions 1 and 2, we have*

(13)
$$\begin{array}{c} ∥{\vec{f}}_{\lambda ,t}-\nabla f_{\rho }+\lambda {\left(L_{K,s}+\lambda I\right)}^{-1}\nabla f_{\rho }{∥}_{K}\leq 2M\frac{s}{\lambda }\kappa c_{\upsilon }c_{h}J_{3}A_{t}. \end{array}$$



**Proof of Lemma** **2.**Taking notice of (Equation 10), it follows that
$$\begin{array}{cc} \; & {\vec{f}}_{\lambda ,t}-\nabla f_{\rho }+\lambda {\left(L_{K,s}+\lambda I\right)}^{-1}\nabla f_{\rho }={\left(L_{K,s}+\lambda I\right)}^{-1}\left({\vec{f}}_{\rho ,s}-L_{K,s}\nabla f_{\rho }\right). \end{array}$$Then, we have
$$\begin{array}{cc} ∥{\vec{f}}_{\lambda ,t}-\nabla f_{\rho }+\lambda {\left(L_{K,s}+\lambda I\right)}^{-1}\nabla f_{\rho }{∥}_{K}\leq & ∥{\left(L_{K,s}+\lambda I\right)}^{-1}{∥}_{K}{∥{\vec{f}}_{\rho ,s}-L_{K,s}\nabla f_{\rho }∥}_{K}\\ \leq & \frac{1}{\lambda }{∥{\vec{f}}_{\rho ,s}-L_{K,s}\nabla f_{\rho }∥}_{K}. \end{array}$$We note that
$$\begin{array}{cc} \; & J_{p}s^{p-2}=\int_{R^{n}}\omega \left(x,\tilde{x}\right)|x-\tilde{x}{|}^{p}d\tilde{x}=\int_{R^{n}}\frac{1}{s^{n+2}}e^{\frac{-{\left|x-\tilde{x}\right|}^{2}}{2s^{2}}}{\left|x-\tilde{x}\right|}^{p}d\;\tilde{x},\;\;\;\;p=2,3,\cdots . \end{array}$$From Assumptions 1 and 2, we have
$$∥{\vec{f}}_{\rho ,s}-L_{K,s}\nabla f_{\rho }{∥}_{K}\leq \int_{Z}\int_{Z}\omega \left(x,\tilde{x}\right){\left|x-\tilde{x}\right|}^{3}\phi \left(z,\tilde{z}\right){∥K_{\tilde{x}}∥}_{K}c_{\upsilon }d\;\rho \left(z\right)d\;\rho \left(\tilde{z}\right)\leq 2Ms\kappa c_{\upsilon }c_{h}J_{3}A_{t}.$$The desired result follows.    □

As for $∥\lambda {\left(L_{K,s}+\lambda I\right)}^{-1}\nabla f_{\rho }{∥}_{\rho }$, the multivariate mean value theorem ensures that there exists $R_{t}\left(\tilde{z}\right)=\phi \left(\tilde{z},\eta_{z}\right),\eta_{z}\in R^{n}\times Y$, such that
(14)$$\begin{array}{cc} \; & \int_{Z}\int_{R^{n}\times Y}e^{-\frac{{\left|x-\tilde{x}\right|}^{2}}{2s^{2}}}\frac{|x-\tilde{x}{|}^{2}}{s^{2+n}}\phi \left(z,\tilde{z}\right)K_{\tilde{x}}\vec{f}\left(\tilde{x}\right)p\left(\tilde{z}\right)d\;zd\;\rho \left(\tilde{z}\right)\\ = & \int_{Z}\int_{R^{n}\times Y}e^{-\frac{{\left|x-\tilde{x}\right|}^{2}}{2s^{2}}}\frac{|x-\tilde{x}{|}^{2}}{s^{2+n}}R_{t}\left(\tilde{z}\right)K_{\tilde{x}}\vec{f}\left(\tilde{x}\right)p\left(\tilde{z}\right)d\;zd\;\rho \left(\tilde{z}\right). \end{array}$$

From (Equation 14), we can define the integral operator associated with the Mercer kernel *K* which is related to $L_{K,s}$. Using Lemma 16 and Lemma 18 in [10], we establish the following Lemma.

**Lemma** **3.**
*Under the Assumption 2, denote $c_{\rho }={\left(2MA_{t}\kappa^{2}c_{h}\left(2J_{2+\varsigma }+J_{4}+c_{h}J_{2}\right)\right)}^{\frac{1}{\varsigma }}$ and $V_{p}=\int_{Z}{\left(p\left(z\right)\right)}^{2}R_{t}\left(z\right)d\;z$. For any $0<s\leq min\{c_{\rho }\lambda^{\frac{1}{\varsigma }},1\}$, we have*

(15)
$$\begin{array}{c} ∥\lambda {\left(L_{K,s}+\lambda I\right)}^{-1}\nabla f_{\rho }{∥}_{\rho }\leq 2\sqrt{\lambda }{\left(V_{p}n{\left(2\pi \right)}^{\frac{n}{2}}M\right)}^{-\frac{1}{2}}{∥L_{K}^{-\frac{1}{2}}\nabla f_{\rho }∥}_{\rho }, \end{array}$$

*where $L_{K}$ is a positive operator on ${\left(L_{\rho_{X}}^{2}\right)}^{n}$ defined by*

$$L_{K}\vec{f}=\int_{Z}K_{x}\vec{f}\left(x\right)\frac{p\left(z\right)R_{t}\left(z\right)}{V_{p}}d\;\rho \left(z\right),\vec{f}\in {\left(L_{\rho }^{2}\right)}^{n}.$$



**Proof of Lemma** **3.**To estimate (Equation 15), we need to consider the convergence of $L_{K,s}$ as s→0. Denote the stepping stone
$$\begin{array}{c} \vec{g}=\int_{Z}\int_{Z}\omega \left(x,\tilde{x}\right)\left(x-\tilde{x}\right)R_{t}\left(\tilde{z}\right)K_{\tilde{x}}{\left(x-\tilde{x}\right)}^{T}\vec{f}\left(\tilde{x}\right)p\left(\tilde{z}\right)d\;zd\;\rho \left(\tilde{z}\right), \end{array}$$
we deduce that
$$\begin{array}{cc} ∥L_{K,s}\vec{f}-2MV_{p}n{\left(2\pi \right)}^{\frac{n}{2}}L_{K}\vec{f}{∥}_{K} & \leq ∥L_{K,s}\vec{f}-\vec{g}+\vec{g}-2MV_{p}n{\left(2\pi \right)}^{\frac{n}{2}}L_{K}\vec{f}{∥}_{K}\\ & \leq ∥L_{K,s}\vec{f}-\vec{g}{∥}_{K}+{∥\vec{g}-2MV_{p}n{\left(2\pi \right)}^{\frac{n}{2}}L_{K}\vec{f}∥}_{K}. \end{array}$$Using the multivariate mean value theorem, there exists $z_{\zeta },z_{\sigma }\in R^{n}\times Y$, such that
$$\begin{array}{cc} ∥L_{K,s}\vec{f}-\vec{g}{∥}_{K}= & ∥p\left(z_{\zeta }\right)\int_{Z}\int_{R^{n}\times Y}R_{t}\left(\tilde{z}\right)\omega \left(x,\tilde{x}\right)\left(\vec{f}{\left(\tilde{x}\right)}^{T}\left(x-\tilde{x}\right)\right)K_{\tilde{x}}\left(x-\tilde{x}\right)d\;zd\;\rho \left(\tilde{z}\right)\\ & -\int_{Z}\int_{Z}\omega \left(x,\tilde{x}\right)\left(x-\tilde{x}\right)R_{t}\left(\tilde{z}\right)K_{\tilde{x}}{\left(x-\tilde{x}\right)}^{T}\vec{f}\left(\tilde{x}\right)p\left(\tilde{z}\right)d\;zd\;\rho \left(\tilde{z}\right){∥}_{K}\\ \leq & ∥p\left(z_{\zeta }\right)\int_{Z}\int_{R^{n}\times Y}R_{t}\left(\tilde{z}\right)\omega \left(x,\tilde{x}\right)\left(\vec{f}{\left(\tilde{x}\right)}^{T}\left(x-\tilde{x}\right)\right)K_{\tilde{x}}\left(x-\tilde{x}\right)d\;zd\;\rho \left(\tilde{z}\right)\\ & -\int_{Z}\int_{R^{n}\times Y}R_{t}\left(\tilde{z}\right)\omega \left(x,\tilde{x}\right)\left(\vec{f}{\left(\tilde{x}\right)}^{T}\left(x-\tilde{x}\right)\right)K_{\tilde{x}}\left(x-\tilde{x}\right)p\left(z\right)d\;zd\;\rho \left(\tilde{z}\right){∥}_{K}\\ & +∥\int_{Z}\int_{Z}R_{t}\left(\tilde{z}\right)\omega \left(x,\tilde{x}\right)\left(\vec{f}{\left(\tilde{x}\right)}^{T}\left(x-\tilde{x}\right)\right)K_{\tilde{x}}\left(x-\tilde{x}\right)\left(p\left(z\right)-p\left(\tilde{z}\right)\right)d\;zd\;\rho \left(\tilde{z}\right){∥}_{K}\\ \leq & ∥p\left(z_{\zeta }\right)-p\left(z_{\sigma }\right)\int_{Z}\int_{R^{n}\times Y}R_{t}\left(\tilde{z}\right)\omega \left(x,\tilde{x}\right)\left(\right)\vec{f}{\left(\tilde{x}\right)}^{T}\left(x-\tilde{x}\right)K_{\tilde{x}}\left(x-\tilde{x}\right)d\;zd\;\rho \left(\tilde{z}\right){∥}_{K}\\ & +∥\int_{Z}\int_{Z}R_{t}\left(\tilde{z}\right)\omega \left(x,\tilde{x}\right)\left(\vec{f}{\left(\tilde{x}\right)}^{T}\left(x-\tilde{x}\right)\right)K_{\tilde{x}}\left(x-\tilde{x}\right)\left(p\left(z\right)-p\left(\tilde{z}\right)\right)d\;zd\;\rho \left(\tilde{z}\right){∥}_{K}\\ \leq & 4Ms^{\varsigma }\kappa c_{h}J_{2+\varsigma }{∥\vec{f}∥}_{\rho }A_{t}. \end{array}$$Noticing $n{\left(2\pi \right)}^{\frac{n}{2}}=J_{2}$, we have
$$\begin{array}{cc} 2V_{p}n{\left(2\pi \right)}^{\frac{n}{2}}ML_{K}\vec{f}= & \int_{Z}\int_{R^{n}\times Y}\omega \left(x,\tilde{x}\right)R_{t}\left(\tilde{z}\right)K_{\tilde{x}}\vec{f}\left(\tilde{x}\right){\left(x-\tilde{x}\right)}^{T}\left(x-\tilde{x}\right)p\left(\tilde{z}\right)d\;zd\;\rho \left(\tilde{z}\right). \end{array}$$Then, by (Equation 7), we can obtain the following conclusion from Lemma 16 in [10] when 0≤s≤1,
$$\begin{array}{cc} ∥\vec{g}-2MV_{p}n{\left(2\pi \right)}^{\frac{n}{2}}L_{K}\vec{f}{∥}_{K} & \leq ∥\int_{Z}\int_{(R^{n}\times Y)\setminus Z}\omega \left(x,\tilde{x}\right)R_{t}\left(\tilde{z}\right)K_{\tilde{x}}\vec{f}\left(\tilde{x}\right)p\left(\tilde{z}\right){\left|x-\tilde{x}\right|}^{2}d\;zd\;\rho \left(\tilde{z}\right){∥}_{K}\\ & \leq 2Mc_{\rho }A_{t}\int_{X}\int_{R^{n}\setminus X}\omega \left(x,\tilde{x}\right)K_{\tilde{x}}|\vec{f}\left(\tilde{x}\right)\left|\right|\left(x-\tilde{x}\right){|}^{2}d\;xd\;\rho_{X}\left(\tilde{x}\right)\\ & \leq 2Ms^{\varsigma }\kappa c_{h}\left(J_{4}+c_{h}J_{2}\right){∥\vec{f}∥}_{\rho }A_{t}. \end{array}$$Combining the above two estimates, there holds for any 0≤s≤1,
(16)$$\begin{array}{c} ∥L_{K,s}-2MV_{p}n{\left(2\pi \right)}^{\frac{n}{2}}L_{K}{∥}_{K}\leq 2MA_{t}\kappa^{2}c_{h}s^{\varsigma }\left(2J_{2+\varsigma }+J_{4}+c_{h}J_{2}\right). \end{array}$$Using Lemma 18 in [10] and (Equation 16), the desired result follows.    □

Since the measure $d\;\tilde{\rho }=\int_{Y}\frac{p\left(z\right)R_{t}\left(z\right)}{V_{p}}d\;\rho$ is probability one on *X*, we know that the operator $L_{K}$ can be used to define the reproducing kernel Hilbert space [22]. Let $L_{K}^{1/2}$ be the $\frac{1}{2}$-th power of the positive operator $L_{K}$ on ${\left(L_{\tilde{\rho }}^{2}\right)}^{n}$ with norm $∥\vec{f}{∥}_{\tilde{\rho }}=(\sum_{l=1}^{n}∥f^{l}{{∥}_{\tilde{\rho }}^{2})}^{1/2}$ having a range in $H_{K}^{n}$, where $∥f^{l}{∥}_{\tilde{\rho }}=(\int_{X}|f^{l}\left(x\right){{|}^{2}d\;\tilde{\rho })}^{1/2}$. Then, $H_{K}^{n}$ is the range of $L_{K}^{1/2}:$(17)$$\begin{array}{c} ∥\vec{f}{∥}_{\tilde{\rho }}={∥L_{K}^{1/2}\vec{f}∥}_{K},\;\;\;\;\vec{f}\in {\left(L_{\tilde{\rho }}^{2}\right)}^{n}. \end{array}$$

The assumption we shall use is $∥L_{K}^{-1/2}\nabla f_{\rho }{∥}_{\tilde{\rho }}<\infty$. It means that $\nabla f_{\rho }$ lies in the range of $L_{K}^{1/2}$. Finally, we can give the upper bound of the error $∥{\vec{f}}_{\lambda ,t}-\nabla f_{\rho }{∥}_{\rho }$.

**Theorem** **2.**
*Under the Assumptions 1 and 2, choose $\lambda =m^{-\frac{\tau }{n+2+3\tau }}$ and $s={\left(\kappa c_{h}\right)}^{\frac{2}{\zeta }}m^{-\frac{1}{n+2+3\tau }}$. For any $m\geq {\left(\kappa c_{h}\right)}^{2(n+2+3\tau )/\tau }$, there exists a constant $C_{\rho ,K}$ such that we have*

(18)
$$∥{\vec{f}}_{\lambda ,t}-\nabla f_{\rho }{∥}_{\rho }\leq C_{\rho ,K}\frac{A_{t}}{\sqrt{B_{t}}}{\left(\frac{1}{m}\right)}^{\frac{\zeta }{2n+4+6\zeta }}.$$



**Proof of Theorem** **2.**Using Cauchy inequality, for $\vec{f}={\left(f^{1},f^{2},\ldots ,f^{n}\right)}^{T}\in {\left(L_{\rho_{X}}^{2}\right)}^{n}$, we have
$$\begin{array}{cc} \int_{X}{\left(f^{l}\left(x\right)\right)}^{2}d\;\rho_{X}\left(x\right) & \leq {\left(\int_{Z}{\left(f^{l}\left(x\right)\right)}^{2}\frac{p\left(z\right)R_{t}\left(z\right)}{V_{p}}d\;\rho \left(z\right)\right)}^{\frac{1}{2}}{\left(\int_{Z}{\left(f^{l}\left(x\right)\right)}^{2}\frac{V_{p}}{p\left(z\right)R_{t}\left(z\right)}d\;\rho \left(z\right)\right)}^{\frac{1}{2}}\\ & \leq \sqrt{\frac{V_{p}}{c_{l}B_{t}}}{\left(\int_{Z}{\left(f^{l}\left(x\right)\right)}^{2}\frac{p\left(z\right)R_{t}\left(z\right)}{V_{p}}d\;\rho \left(z\right)\right)}^{\frac{1}{2}}{\left(\int_{X}{\left(f^{l}\left(x\right)\right)}^{2}d\;\rho_{X}\left(x\right)\right)}^{\frac{1}{2}}. \end{array}$$It means that
$$\begin{array}{c} {\left(\int_{X}{\left(f^{l}\left(x\right)\right)}^{2}d\;\rho_{X}\left(x\right)\right)}^{\frac{1}{2}}\leq \sqrt{\frac{V_{p}}{c_{l}B_{t}}}{\left(\int_{Z}{\left(f^{l}\left(x\right)\right)}^{2}\frac{p\left(z\right)R_{t}\left(z\right)}{V_{p}}d\;\rho \left(z\right)\right)}^{\frac{1}{2}}. \end{array}$$According to the definitions of $∥f^{l}{∥}_{\rho }$ and $∥f^{l}{∥}_{\tilde{\rho }}$, it is trivial to obtain
(19)$$\begin{array}{c} ∥\vec{f}{∥}_{\rho }\leq \sqrt{\frac{V_{p}}{c_{l}B_{t}}}{∥\vec{f}∥}_{\tilde{\rho }}. \end{array}$$Since $s={\left(\kappa c_{h}\right)}^{\frac{2}{\zeta }}\lambda^{\frac{1}{\zeta }},\lambda ={\left(\frac{1}{m}\right)}^{\frac{\zeta }{n+2+3\zeta }}$, we see from the fact $J_{2}>1$ that the restriction $0<s\leq \min\{c_{\rho }\lambda^{\frac{1}{\zeta }},1\}$ in Lemma 3 is satisfied for $m\geq {\left(\kappa c_{h}\right)}^{2(n+2+3\tau )/\tau }$. Then, combining Lemmas 2 and 3, Equation (Equation 17) and inequality (Equation 19), we have
$$\begin{array}{cc} ∥{\vec{f}}_{\lambda ,t}-\nabla f_{\rho }{∥}_{\rho }\leq & ∥{\vec{f}}_{\lambda ,t}-\nabla f_{\rho }+\lambda {\left(L_{K,s}+\lambda I\right)}^{-1}\nabla f_{\rho }{∥}_{\rho }+{∥\lambda {\left(L_{K,s}+\lambda I\right)}^{-1}\nabla f_{\rho }∥}_{\rho }\\ \leq & \kappa ∥{\vec{f}}_{\lambda ,t}-\nabla f_{\rho }+\lambda {\left(L_{K,s}+\lambda I\right)}^{-1}\nabla f_{\rho }{∥}_{K}+{∥\lambda {\left(L_{K,s}+\lambda I\right)}^{-1}\nabla f_{\rho }∥}_{\rho }\\ \leq & 2M\frac{s}{\lambda }\kappa^{2}c_{\upsilon }c_{h}J_{3}A_{t}+2\sqrt{\frac{V_{p}}{c_{l}B_{t}}}\sqrt{\lambda }{\left(MV_{p}n{\left(2\pi \right)}^{\frac{n}{2}}\right)}^{-\frac{1}{2}}{∥\nabla f_{\rho }∥}_{K}\\ \leq & C_{\rho ,K}\frac{A_{t}}{\sqrt{B_{t}}}{\left(\frac{1}{m}\right)}^{\frac{\zeta }{2n+4+6\zeta }}, \end{array}$$
where $C_{\rho ,K}=\left({\left(2\kappa c_{h}\right)}^{\frac{2}{\zeta }}+2\right)\max\left\{M\kappa^{2}c_{\upsilon }c_{h}J_{3},\sqrt{\frac{V_{p}}{c_{l}}}{\left(MV_{p}n{\left(2\pi \right)}^{\frac{n}{2}}\right)}^{-\frac{1}{2}}C_{K}\right\}$.    □

**Remark** **2.**
*Theorem 2 shows when m→+∞, $∥{\vec{f}}_{\lambda ,t}-\nabla f_{\rho }{∥}_{\rho }\to 0$. This means that the scheme (Equation 5) is consistent. In addition, $A_{t}$ and $B_{t}$ tend to 1 as t tends 0, we can see that the convergence rate of Scheme (Equation 5) is $-\frac{\zeta }{2n+4+6\zeta }$, which is consistent with previous result in [10]. It means that the proposed method can be regarded as an extension of traditional GL.*


## 3. Computing Algorithm

In this section, we present the GL model under TERM and propose to use the gradient descent algorithm to find the minimizer. Finally, the convergence of the proposed algorithm is also guaranteed.

Given a set of observations $z={\left\{z_{i}=\left(x_{i},y_{i}\right)\right\}}_{i=1}^{m}\in Z^{m}$ independently drawn according to ρ and assume that the RKHS are rich that the kernel matrix $K={\left(K\left(x_{i},x_{j}\right)\right)}_{i,j=1}^{m}$ is strictly positive definite [27]. According to the Representer Theorem of kernel methods [28], we assert the approximation of ${\vec{f}}_{\lambda ,t}$ has the following form: $\sum_{i=1}^{m}c_{i}K_{x_{i}},c_{i}={\left(c_{i}^{1},\ldots ,c_{i}^{n}\right)}^{T}\in R^{n}.$ Let $c={\left(c_{1}^{T},\cdots ,c_{m}^{T}\right)}^{T}\in R^{mn}$, the empirical version of (Equation 4) is formulated as follows:(20)$$\begin{array}{c} c_{z,\lambda }:=\underset{c\in R^{mn}}{\arg\min}\left\{E_{z}\left(c,t\right)+\lambda ∥\sum_{i=1}^{m}c_{i}K_{x_{i}}{∥}_{K}^{2}\right\}, \end{array}$$
where
$$E_{z}\left(c,t\right)=\frac{1}{t}\log\left(\frac{1}{m^{2}}\sum_{i,j=1}^{m}\exp\left\{t\omega \left(x_{i},x_{j}\right){\left(y_{i}-y_{j}+\sum_{p=1}^{m}K\left(x_{p},x_{i}\right){\hat{x}}_{ij}c_{p}\right)}^{2}\right\}\right),$$
with ${\hat{x}}_{ij}={\left(x_{j}-x_{i}\right)}^{T}$. For simplicity, we denote
$$V_{z}\left(c,z_{i},z_{j}\right)=\omega \left(x_{i},x_{j}\right){\left(y_{i}-y_{j}+\sum_{p=1}^{m}K\left(x_{p},x_{i}\right){\hat{x}}_{ij}c_{p}\right)}^{2}$$
and
$$\phi_{z}\left(c,z_{i},z_{j}\right)=\exp\left\{t\left(V_{z}\left(c,z_{i},z_{j}\right)-E_{z}\left(c,t\right)\right)\right\}.$$

The gradients of $E_{z}\left(c,t\right)$ and $∥\sum_{i=1}^{m}c_{i}K_{x_{i}}{∥}_{K}^{2}$ at *c* are given by
$$\begin{array}{cc} \nabla_{c}E_{z}\left(c,t\right)= & \frac{1}{m^{2}}\sum_{i,j=1}^{m}\phi_{z}\left(c,z_{i},z_{j}\right)2\omega \left(x_{i},x_{j}\right)\left(y_{i}-y_{j}+\sum_{p=1}^{m}K\left(x_{p},x_{i}\right){\hat{x}}_{ij}c_{p}\right)\times \\ & {\left(K\left(x_{1},x_{i}\right){\hat{x}}_{ij},\cdots ,K\left(x_{m},x_{i}\right){\hat{x}}_{ij}\right)}^{T}, \end{array}$$
and
$$\nabla_{c}∥\sum_{i=1}^{m}c_{i}K_{x_{i}}{∥}_{K}^{2}=2\sum_{i=1}^{m}{\left(K\left(x_{i},x_{1}\right)c_{i}^{T},\cdots ,K\left(x_{i},x_{m}\right)c_{i}^{T}\right)}^{T}.$$

Correspondingly, scheme (Equation 20) can be solved via the following gradient method:(21)$$c^{k}=c^{k-1}-\alpha \left(\nabla_{c}E_{z}\left(c^{k-1},t\right)+\lambda \nabla_{c}∥\sum_{i=1}^{m}c_{i,k-1}K_{x_{i}}{∥}_{K}^{2}\right),$$
where $c^{k}={\left(c_{1,k}^{T},\cdots ,c_{m,k}^{T}\right)}^{T}\in R^{mn}$ is the calculated solution at iteration *k*, and α is the step-size. The detailed gradient descent scheme is stated in Algorithm 1. To prove the convergence, we introduce the following lemma derived from Theorem 1 in [29].

**Lemma** **4.**
*When h(c) has an γ-Lipschitz continuous gradient (γ-smoothness) and is μ-strongly convex, for the basic unconstrained optimization problem $c^{*}=\arg\min h\left(c\right)$, the gradient descent algorithm $c^{k}=c^{k-1}-\frac{1}{\gamma }\nabla h\left(c^{k-1}\right)$ with a step-size of 1/γ has a global linear convergence rate*

$$h\left(c^{k}\right)-h\left(c^{*}\right)\leq {\left(1-\frac{\mu }{\gamma }\right)}^{k}\left(h\left(c^{0}\right)-h\left(c^{*}\right)\right).$$



**Algorithm 1** Gradient descent for the Gradient Learning under TERM

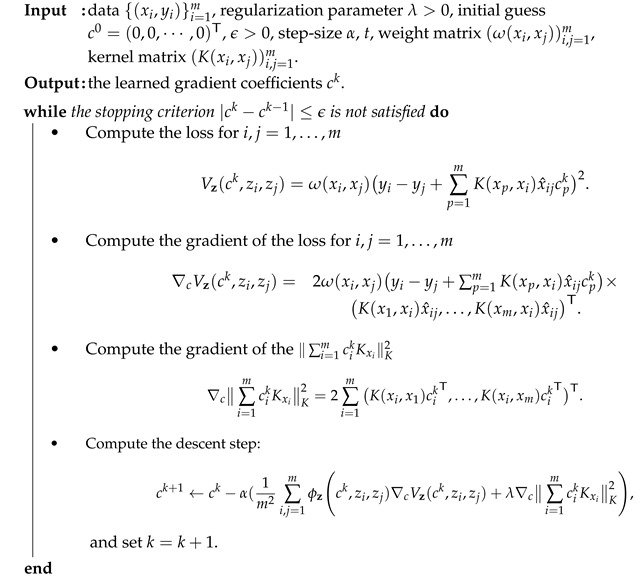



From Lemma 4, we obtain the following conclusion which states that the proposed algorithm converges to (Equation 20) by choosing a suitable step size α.

**Theorem** **3.**
*Denote $L\left(c,t\right)=E_{z}\left(c,t\right)+\lambda {∥\sum_{i=1}^{m}c_{i}K_{x_{i}}∥}_{K}^{2}$, $\beta_{max},\beta_{min}$ are the maximum and minimum eigenvalues of kernel matrix K, respectively. There exist $\mu \in R^{+}$ and $\gamma \in R^{+}$ dependent on t such that $L(c^{k},t)$ is γ-smoothness and μ-strongly convex for any $t>(-n\lambda \beta_{min}/64\left(M^{2}+C_{K}M_{X}\right)M_{X}^{2}m\kappa^{4})$. In addition, let the minimizer $c_{z,\lambda }$ defined in scheme (Equation 20) and $\left\{c^{k}\right\}$ be the sequence generated by Algorithm 1 with α=1/γ, we have*

(22)
$$\begin{array}{c} L\left(c^{k},t\right)-L\left(c_{z,\lambda },t\right)\leq {\left(1-\frac{\mu }{\gamma }\right)}^{k}\left(L\left(c^{0},t\right)-L\left(c_{z,\lambda },t\right)\right). \end{array}$$



**Proof of Theorem** **3.**Note that the strong convexity and the smoothness are related to the Hessian Matrix, and we provide the proof by dividing the Hessian Matrix into three parts:
(23)$$\begin{array}{cc} \nabla_{cc^{T}}^{2}L\left(c,t\right)= & \underset{E_{1}}{\underset{︸}{\frac{t}{m^{2}}\sum_{i,j=1}^{m}\phi_{z}\left(c,z_{i},z_{j}\right)\left(\nabla_{c}V_{z}\left(c,z_{i},z_{j}\right)-\nabla_{c}E_{z}\left(c,t\right)\right)\nabla_{c}V_{z}{\left(c,z_{i},z_{j}\right)}^{T}}}\\ & +\underset{E_{2}}{\underset{︸}{\frac{1}{m^{2}}\sum_{i,j=1}^{m}\phi_{z}\left(c,z_{i},z_{j}\right)\nabla_{cc^{T}}^{2}V_{z}\left(c,z_{i},z_{j}\right)}}+\underset{E_{3}}{\underset{︸}{\lambda \nabla_{cc^{T}}^{2}{∥\sum_{i=1}^{m}c_{i}K_{x_{i}}∥}_{K}^{2}}}. \end{array}$$*(1) Estimation on $E_{1}$:* Note that $m^{2}\nabla_{c}E_{z}\left(c,t\right)=\sum_{i,j=1}^{m}\phi_{z}\left(c,z_{i},z_{j}\right)\nabla_{c}V_{z}\left(c,z_{i},z_{j}\right)$ and $\sum_{i,j=1}^{m}\phi_{z}\left(c,z_{i},z_{j}\right)=m^{2}$. It follows that
$$\begin{array}{c} \sum_{i,j=1}^{m}\phi_{z}\left(c,z_{i},z_{j}\right)\left(\nabla_{c}V_{z}\left(c,z_{i},z_{j}\right)-\nabla_{c}E_{z}\left(c,t\right)\right)\nabla_{c}^{T}E_{z}\left(c,t\right)=0. \end{array}$$Hence, we can get the following equation:
(24)$$\begin{array}{c} E_{1}=\frac{t}{m^{2}}\sum_{i,j=1}^{m}\phi_{z}\left(c,z_{i},z_{j}\right)\left(\nabla_{c}V_{z}\left(c,z_{i},z_{j}\right)-\nabla_{c}E_{z}\left(c,t\right)\right){\left(\nabla_{c}V_{z}\left(c,z_{i},z_{j}\right)-\nabla_{c}E_{z}\left(c,t\right)\right)}^{T}. \end{array}$$Similar to the proof of Lemma 1, for i,j=1,⋯,m, it directly follows that
$$\begin{array}{c} \omega \left(x_{i},x_{j}\right)\left(y_{i}-y_{j}+\sum_{p=1}^{m}K\left(x_{p},x_{i}\right){\hat{x}}_{ij}c_{p}\right)\leq 2\sqrt{2(M^{2}+C_{K}M_{X})}. \end{array}$$Note that, for i,j=1,⋯,m, $\nabla_{c}V_{z}\left(c,z_{i},z_{j}\right)\nabla_{c}V_{z}{\left(c,z_{i},z_{j}\right)}^{T}$ has a sole eigenvalue, it means
(25)$$\begin{array}{cc} \nabla_{c}V_{z}\left(c,z_{i},z_{j}\right)\nabla_{c}V_{z}{\left(c,z_{i},z_{j}\right)}^{T}⪯ & 32\left(M^{2}+C_{K}M_{X}\right)M_{X}^{2}m\kappa^{4}I_{mn}, \end{array}$$
and we have
$$\begin{array}{c} {\left(\nabla_{c}V_{z}\left(c,z_{i},z_{j}\right)-\nabla_{c}E_{z}\left(c,t\right)\right)}^{T}\left(\nabla_{c}V_{z}\left(c,z_{i},z_{j}\right)-\nabla_{c}E_{z}\left(c,t\right)\right)\leq 128\left(M^{2}+C_{K}M_{X}\right)M_{X}^{2}m\kappa^{4}. \end{array}$$It means that the maximum eigenvalue of $E_{1}$ is $128t\left(M^{2}+C_{K}M_{X}\right)M_{X}^{2}m\kappa^{4}$. Then, the following inequations are satisfied
(26)$$\begin{array}{c} \left\{\begin{array}{c} 0_{mn}⪯E_{1}⪯128t\left(M^{2}+C_{K}M_{X}\right)M_{X}^{2}m\kappa^{4}I_{mn},\;\;\;t>0;\\ 128t\left(M^{2}+C_{K}M_{X}\right)M_{X}^{2}m\kappa^{4}I_{mn}⪯E_{1}⪯0_{mn},\;\;\;t<0, \end{array}\right. \end{array}$$
where $0_{mn}$ is the mn×mn matrix with all elements zero.*(2) Estimation on $E_{2}$:* Note that $\nabla_{cc^{T}}^{2}V_{z}\left(c,z_{i},z_{j}\right)$ can be rewritten as
$$\begin{array}{c} 2\omega \left(x_{i},x_{j}\right)\left(K\left(x_{1},x_{i}\right){\hat{x}}_{ij},\cdots ,K\left(x_{m},x_{i}\right){\hat{x}}_{ij}\right){\left(K\left(x_{1},x_{i}\right){\hat{x}}_{ij},\cdots ,K\left(x_{m},x_{i}\right){\hat{x}}_{ij}\right)}^{T}. \end{array}$$Similar to (Equation 25), we have $\nabla_{cc^{T}}^{2}V_{z}\left(c,z_{i},z_{j}\right)⪯2\kappa^{4}M_{x}^{2}I_{mn}$. It follows
(27)$$\begin{array}{c} 0_{mn}⪯E_{2}⪯2\kappa^{4}M_{x}^{2}I_{mn}. \end{array}$$*(3) Estimation on $E_{3}$:* By a direct computation, we have
$$\begin{array}{c} E_{3}=2\lambda \left[\begin{array}{cccc} I_{n}K\left(x_{1},x_{1}\right) & I_{n}K\left(x_{1},x_{2}\right) & \cdots & I_{n}K\left(x_{1},x_{m}\right)\\ I_{n}K\left(x_{2},x_{1}\right) & I_{n}K\left(x_{2},x_{2}\right) & \cdots & I_{n}K\left(x_{2},x_{m}\right)\\ \vdots & \vdots & \ddots & \vdots \\ I_{n}K\left(x_{m},x_{1}\right) & I_{n}K\left(x_{m},x_{2}\right) & \cdots & I_{n}K\left(x_{m},x_{m}\right) \end{array}\right]. \end{array}$$Setting $Q={\left(q_{11},q_{21},\cdots ,q_{n1},\cdots ,q_{1m},q_{2m},\cdots ,q_{nm}\right)}^{T}\in R^{mn}$, we deduce that
$$\begin{array}{c} Q^{T}E_{3}Q=2\lambda \sum_{l=1}^{n}\sum_{i=1}^{m}\sum_{j=1}^{m}K\left(x_{i},x_{j}\right)q_{li}q_{lj}. \end{array}$$Note that the matrix of quadratic form $\sum_{i=1}^{m}\sum_{j=1}^{m}K\left(x_{i},x_{j}\right)q_{li}q_{lj}$ is K, then we can obtain
(28)$$\begin{array}{c} 2\lambda n\beta_{min}I_{mn}⪯E_{3}⪯2\lambda n\beta_{max}I_{mn}. \end{array}$$Combining (Equation 26), (Equation 27) and (Equation 28), there exist two constants
$$\mu =\min\{2n\lambda \beta_{min}+128t\left(M^{2}+C_{K}M_{X}\right)M_{X}^{2}m\kappa^{4},2n\lambda \beta_{min}\}$$
and
$$\gamma =\max\{128t\left(M^{2}+C_{K}M_{X}\right)M_{X}^{2}m\kappa^{4}+2n\lambda \beta_{max},2\kappa^{4}M_{x}^{2}+2n\lambda \beta_{max}\}$$
satisfying that
$$\begin{array}{c} \mu I_{mn}⪯\nabla_{cc^{T}}^{2}L\left(c,t\right)⪯\gamma I_{mn}. \end{array}$$Note μ>0 as $t>-n\lambda \beta_{min}/64\left(M^{2}+C_{K}M_{X}\right)M_{X}^{2}m\kappa^{4}$, and it means that L(c,t) is γ-smoothness and μ-strongly convex. The desired result follows by Lemma 4. □

## 4. Simulation Experiments

In this section, we carry out simulation studies with the TGL model (t<0 for robust) on a synthetic data set in the robust variable selection problem. Let the observation data set $z={\left\{z_{i}=\left(x_{i},y_{i}\right)\right\}}_{i=1}^{m}$ with $x_{i}=\left(x_{i}^{1},\cdots ,x_{i}^{n}\right)$ be generated by the following linear equations:$$y_{i}=x_{i}\cdot w+\epsilon ,$$
where ϵ represents the outliers or noises. To be specific, three different noises are used: Cauchy noise with the location parameter a=2 and scale parameter b=4, Chi-square noise with 5 DOF scaled by 0.01 and Gaussian noise N(0,0.3). Three different proportions of outliers including 0%, 20%, or 40% are drawn from the Gaussian noise N(0,100). Meanwhile, we consider two different cases with (m,n)=(50,50),(30,80) corresponding to *m* = *n* and *m* < *n*, respectively. The weighted vector $w=(w^{1},\cdots ,w^{n})$ over different dimensions is constructed as follows:

$w^{l}=2+0.5\sin\left(\frac{2\pi l}{10}\right)$, for $l=1,\cdots ,N_{n}$ and 0, otherwise.

Here, $N_{n}=30$ means the number of effective variables. Two situations including uncorrelated variables $x∼N(0_{n},I_{n})$ and correlated variables $x∼N(0_{n},\Sigma_{n})$ are implemented for *x*, where the covariance matrix $\Sigma_{n}$ is given with the (l,p)th entry $0.5^{\left|l-p\right|}$.

For the variable selection algorithms, we perform the TGL with $t=6\times 10^{-6},-1,-10$ and compare the traditional GL model [10] and RGL model [20]. For the GL and TGL models, $N_{n}$ variables are selected by ranking
$$\begin{array}{c} r_{l}=\frac{∥f_{z,\lambda }^{l}{∥}_{K}^{2}}{\sum_{p=1}^{n}{∥f_{z,\lambda }^{p}∥}_{K}^{2}},\;\;\;l=1,\cdots ,n. \end{array}$$

For the RGL model, $N_{n}$ variables are selected by ranking
$$\begin{array}{c} r_{l}=\frac{\sum_{i=1}^{m}{\left(c_{i}^{l}\right)}^{2}}{\sum_{q=1}^{n}\sum_{i=1}^{m}{\left(c_{i}^{q}\right)}^{2}},\;\;\;l=1,\cdots ,n. \end{array}$$

A model selecting more effective variables $\left(\leq N_{n}\right)$ means a better algorithm.

We repeat experiments for 30 times with the observation set z generated in each circumstance. The average selected effective variables for different circumstances are reported in Table 1, and the optimal results are marked in bold. Several useful conclusions can be drawn from Table 1.

(1) When the input variables are uncorrelated, the three models have similar performance under different noise conditions and can provide satisfactory variable selection results (approaching $N_{n}$) without outliers. However, the performance degrades severely for GL and a little for TGL (t<0 for robust) with the increasing proportions of outliers, especially in case (m,n)=(30,80). In contrast, RGL can always provide satisfying performance. This is consistent with the previous phenomenon [20].

(2) When the input variables are correlated, the three models also have similar performance under different noise conditions but only can select partial effective variables ranging from $N_{n}/3$ to $2N_{n}/3$. In general, they degrade slowly with the increasing proportions of outliers and perform better in case (m,n)=(50,50) than in (m,n)=(30,80). Specifically, the TGL model with t=−1 gives slightly better selection results than GL and RGL in case (m,n)=(50,50). It supports the superiority of TGL to some extent.

(3) It is worth noting that the TGL model with $t=6\times 10^{-6}$ has similar performance to GL. This phenomenon supports the theoretical conclusion that TGL recovers the GL as t→0 and the algorithmic effectiveness that the proposed gradient descent method can converge to the minimizer.

(4) Noting that the TGL model with different parameters *t* has great differences in the variable selection results, we further conduct some simulation studies to investigate the influence. Figure 1 shows the variable selection results of different parameters *t* ranging from −100 to −0.1. We can see that the satisfying performance can be achieved when the parameter *t* is near −1. It does not turn out well when |t| is too large. This coincides with our previous discussion that L(c,t) is strongly convex with limited *t*.

## 5. Conclusions

In this paper, we have proposed a new learning objective TGL by embedding the *t*-tilted loss into the GL model. On the theoretical side, we have established its consistency and provided the convergence rate with the help of error decomposition and operator approximation technique. On the practical side, we have proposed a gradient descent method to solve the learning objective and provided the convergence analysis. Simulated experiments have verified the theoretical conclusion that TGL recovers the GL as t→0 and the algorithmic effectiveness that the proposed gradient descent method can converge to the minimizer. In addition, they also demonstrated the superiority of TGL when the input variables are correlated. Along the line of the present work, several open problems deserve further research—for example, using the random feature approximation to scale up the kernel methods [30] and learning with data-dependent hypothesis space to achieve a tighter error bound [31]. These problems are under our research.

## Figures and Tables

**Figure 1 entropy-24-00956-f001:**
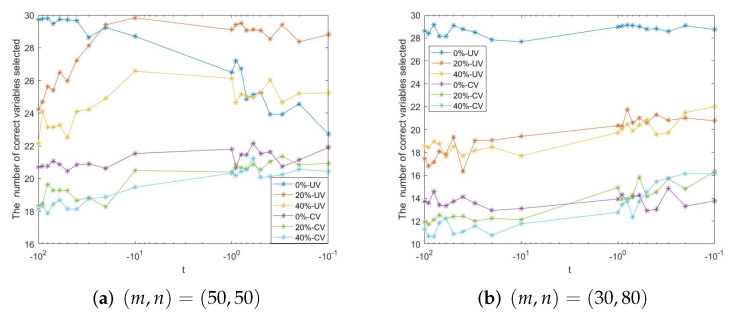
The influence of different *t* on the variable selection results.

**Table 1 entropy-24-00956-t001:** Variable selection results for different circumstances.

	Methods	Uncorrelated Variables			Correlated Variables		
		0%	20%	40%	0%	20%	40%
Cauchy noise	GL	28.70	24.27	19.03	20.27	17.53	16.53
(m,n)=(50,50)	RGL	29.00	26.57	27.7	20.80	15.40	14.16
	$TGL_{t=6\times 10^{-6}}$	29.63	24.06	18.04	20.67	17.00	16.23
	$TGL_{t=-1}$	29.53	26.07	26.00	21.07	17.6	17.13
	$TGL_{t=-10}$	29.53	24.23	24.03	16.93	15.78	15.67
Chi-square noise	GL	29.40	24.73	20.37	18.40	17.93	16.03
(m,n)=(50,50)	RGL	29.63	26.90	27.60	19.90	16.10	14.67
	$TGL_{t=6\times 10^{-6}}$	29.84	24.4	20.90	18.20	17.30	17.20
	$TGL_{t=-1}$	29.14	24.56	25.18	21.10	18.77	17.93
	$TGL_{t=-10}$	25.13	24.10	24.93	20.83	17.10	16.60
Gaussian noise	GL	28.83	25.16	20.13	18.04	16.70	15.93
(m,n)=(50,50)	RGL	29.40	26.70	27.20	19.87	16.40	14.36
	$TGL_{t=6\times 10^{-6}}$	29.23	25.23	20.20	18.37	17.76	16.3
	$TGL_{t=-1}$	27.63	26.20	25.90	21.06	18.40	17.90
	$TGL_{t=-10}$	22.9	25.23	25.06	21.43	17.13	16.23
Cauchy noise	GL	29.60	11.33	12.30	11.93	11.57	10.97
(m,n)=(30,80)	RGL	29.87	29.97	29.93	16.50	16.97	15.20
	$TGL_{t=6\times 10^{-6}}$	28.47	10.67	10.49	11.13	11.03	10.93
	$TGL_{t=-1}$	27.06	20.67	11.3	17.08	14.4	11.56
	$TGL_{t=-10}$	16.66	16.23	15.12	13.97	13.92	13.54
Chi-square noise	GL	29.83	11.47	12.57	12.57	11.67	11.33
(m,n)=(30,80)	RGL	29.93	29.93	29.71	19.87	18.80	17.50
	$TGL_{t=6\times 10^{-6}}$	29.03	11.10	12.90	12.50	10.87	11.43
	$TGL_{t=-1}$	29.37	23.60	23.53	16.08	14.4	11.40
	$TGL_{t=-10}$	28.17	23.33	23.23	13.97	13.92	13.54
Gaussian noise	GL	29.77	11.83	12.27	12.92	12.44	11.54
(m,n)=(30,80)	RGL	29.70	29.93	29.93	19.73	13.67	9.83
	$TGL_{t=6\times 10^{-6}}$	28.47	10.67	10.49	13.06	9.79	8.73
	$TGL_{t=-1}$	27.06	20.67	11.3	16.08	14.4	11.90
	$TGL_{t=-10}$	16.66	16.23	15.12	13.97	13.92	13.54

## Data Availability

Not applicable.

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
