# Peer review of "Gradient Learning under Tilted Empirical Risk Minimization"

_entropy, 2022, doi:10.3390/e24070956_

Round 1
Reviewer 1 Report
- Problem formulation seems sufficient on the point of view. Therefore, the part of "Theoretical guarantees" shall be mentioned more analysis procedure in order to emphasize the readers.
- Under the proof of Theorem 2, can author provide more details of the inequality (18) according the equation above line number 108 page 7.
- The specify of the learning equation (20) altogether with the descent step in Algorithm 1 should be explained for some details altogether with the learning rate \alpha.
- In general, the learning rate may play an importance role of the convergence property of the gradient search. In this work, how this effect regrading to the proposed scheme.
- The numerical results seem sufficient but the conclusion may be required a bit revision in order to emphasize the main contributions mentioned in the introduction section.
Reviewer 2 Report
Here authors have given a new learning objective TGL and established its theoretical support from the function approximation viewpoint. They have proposed a gradient descent method and provided the convergence analysis. Simulated experimental results demonstrate that the proposed TGL model can provide slightly better selection results than other GL models when the input variables are correlated. Along the line of the present work, several open problems deserve further research. For example, using the random feature approximation to scale up the kernel methods [Scalable Kernel Methods via Doubly Stochastic Gradients. In Proceedings of the Advances in Neural Information Processing Systems; Ghahramani, Z.;Welling, M.; Cortes, C.; Lawrence, N.;Weinberger, K., Eds. Curran Associates, Inc., 2014, Vol. 27.] and learning with data-dependent hypothesis space to achieve a tighter error bound [Regularized modal regression with data-dependent hypothesis spaces. International Journal of Wavelets, Multiresolution and Information Processing 2019, 17, 1950047.].
In my opinion these results are quite interesting and significant and hence can be accepted for publication.
I suggest that the authors should give a proper and clearer distinction and edge of new GL model over old one.
